# Bézier Gaussian Processes for Tall and Wide Data

**Martin Jørgensen**
Department of Engineering Science
University of Oxford
`martinj@robots.ox.ac.uk`

**Michael A. Osborne**
Department of Engineering Science
University of Oxford
`mosb@robots.ox.ac.uk`

## Abstract

Modern approximations to Gaussian processes are suitable for "tall data", with a cost that scales well in the number of observations, but under-performs on "wide data", scaling poorly in the number of input features. That is, as the number of input features grows, good predictive performance requires the number of summarising variables, and their associated cost, to grow rapidly. We introduce a kernel that allows the number of summarising variables to grow exponentially with the number of input features, but requires only linear cost in both number of observations and input features. This scaling is achieved through our introduction of the *Bézier buttress*, which allows approximate inference without computing matrix inverses or determinants. We show that our kernel has close similarities to some of the most used kernels in Gaussian process regression, and empirically demonstrate the kernel's ability to scale to both tall and wide datasets.

Gaussian processes (GPs) are a probabilistic approach to modelling functions that permit tractable Bayesian inference. They are, however, notorious for their poor scalability. In recent decades, this criticism has been challenged. Several approximate methods now allow GPs to scale to millions of data points. Yet, scalability in the number of data points is merely one challenge of big data. There are still problems associated with the *input* dimensionality – one aspect of the famed *curse of dimensionality*. Burt et al. [2020] analysed the most studied approximation, the so-called sparse inducing points methods, and showed it to be accurate for low dimensional inputs. Alarmingly, exponentially many inducing points are still needed in high-dimensional input spaces, that is, for problems with a large number of features. As such, despite modern GP approximations scaling to *tall* data, they are still discounted when concerning *wide* data.

In response to this, there exist GP approximations built on simplices or grid-structures in the input space [Wilson and Nickisch, 2015, Gardner et al., 2018, Kapoor et al., 2021]. These take advantage of attractive fast linear algebra, but are often limited by memory in higher dimensions. Their advantage is the ability to fill the input space with structured points, so all observations have a close neighbour.

We propose a new kernel for GP regression that requires neither matrix inversion nor determinant calculation – GPs' two core computational sinners. Additionally, we cover the input space[1] with exponentially many points, but introduce an approximation that grows only linearly in computational complexity. That is, our method scales linearly in both the number of data points *and* the number of input dimensions, whilst being space-filling in the input domain.

GPs are indispensable to fields where uncertainty is a driver in decision-making mechanisms. Such fields include Bayesian optimisation, active learning and reinforcement learning. The critical decision mechanism is the exploration-exploitation trade-off. One ability useful in such fields is to assign high uncertainty to unexplored regions, just as does an exact GP. We show that our proposed model also assigns high uncertainty to unexplored regions, suggesting our model as well-suited to decision-making problems.

---

[1]A limiting assumption of our kernel is its restriction to a box-bounded domain in the input space.

36th Conference on Neural Information Processing Systems (NeurIPS 2022).

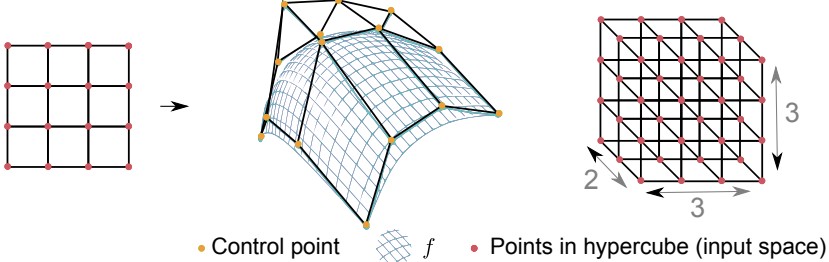

Control point    $f$    Points in hypercube (input space)

Figure 1: *Left*: Illustration of 3d control points (orange) and the 2d grid spanning the input space $[0, 1] \times [0, 1]$. We see how the function (or surface) $f$ gets impacted by the control points. *Right*: How the points cover the hypercube in 3 dimensions. Here a grid in 3d with orders $(2, 3, 3)$.

## 1   Background

**Bézier curves and surfaces** are parametrised geometric objects that have found great usage in computer-aided design and robotics [Prautzsch et al., 2002]. The simplest Bézier curve is the linear interpolation of two points $\boldsymbol{p}_0$ and $\boldsymbol{p}_1$ in $\mathbb{R}^D$; the Bézier curve of order 1

$$\boldsymbol{c}(t) = (1 - t)\boldsymbol{p}_0 + t\boldsymbol{p}_1, \quad t \in [0, 1]. \tag{1}$$

Higher order curves are generalised in the following way: the order-$\nu$ Bézier curve is defined as

$$\boldsymbol{c}(t) = \sum_{i=0}^{\nu} B_i^{\nu}(t)\boldsymbol{p}_i, \quad t \in [0, 1]. \tag{2}$$

In Bézier terms, $\boldsymbol{p}_i$ are referred to as *control points*. Notice an order-$\nu$ Bézier curve has $\nu + 1$ control points. $B_i^{\nu}$ denotes the $i$th Bernstein polynomial of order $\nu$. They are defined as

$$B_i^{\nu}(t) = \frac{\nu!}{i!(\nu - i)!} t^i (1 - t)^{\nu - i}. \tag{3}$$

By going from curves to *surfaces* we wish to extend from the scalar $t$ to a spatial input $\boldsymbol{x} \in [0, 1]^d$, for $d > 1$. Here, we can define Bézier $d$-surfaces as

$$\boldsymbol{c}_d(\boldsymbol{x}) = \sum_{i_1=0}^{\nu_1} \sum_{i_2=0}^{\nu_2} \cdots \sum_{i_d=0}^{\nu_d} B_{i_1}^{\nu_1}(x_1) \cdots B_{i_d}^{\nu_d}(x_d) \boldsymbol{p}_{i_1,\ldots,i_d}, \quad \boldsymbol{x} = (x_1, x_2, \ldots, x_d) \in [0, 1]^d. \tag{4}$$

Figure 1 gives a visual illustration of a 2-dimensional surface embedded in $\mathbb{R}^3$. In the literature, it is difficult to find any studies of $d$-surfaces for $d > 2$. This paper targets especially this high-dimensional input case. We restrict our output dimension to 1, the regression problem, but the methods naturally extend to multidimensional outputs. The red points in Figure 1 show how each control points has an associated location in the input space. They are placed on a grid-like structure, and the order of each dimension determines how fine the mesh-grid of the hypercube is; i.e. how dense the input-space is filled.

**Gaussian processes** (GPs) are meticulously studied in probability and statistics [Williams and Rasmussen, 2006]. They provide a way to define a probability distribution over functions. This makes them useful, as priors, to build structures for quantifying uncertainty in prediction. They are defined as a probability measure over functions $f : \mathcal{X} \to \mathbb{R}$, such that any collection of elements $(\boldsymbol{x}_1, \boldsymbol{x}_2, \ldots, \boldsymbol{x}_n)$ in $\mathcal{X}$ have their associated output $(f(\boldsymbol{x}_1), \ldots, f(\boldsymbol{x}_n))$ following a joint Gaussian distribution. This distribution is fully determined by a mean function $m : \mathcal{X} \to \mathbb{R}$ and a positive semi-definite kernel function $k : \mathcal{X} \times \mathcal{X} \to \mathbb{R}$. They admit exact Bayesian inference. However, exact inference comes at a prohibitive worst-case computational cost of $\mathcal{O}(n^3)$, where $n$ is the number of training points, due to computing inverse and determinant of the kernel matrix.

Sparse Gaussian processes [Snelson and Ghahramani, 2005] overcome this burden by conditioning on $m$ inducing points, reducing complexity to $\mathcal{O}(nm^2)$, where usually $m \ll n$. The inducing points, denoted $\boldsymbol{u}$, are then marginalised to obtain an approximate posterior of $f$. The variational posterior mean and variance, at a location $\boldsymbol{x}^*$ are then given by

$$\mathbb{E}[f(\boldsymbol{x}^*)] = k(\boldsymbol{x}^*, \boldsymbol{Z})k(\boldsymbol{Z}, \boldsymbol{Z})^{-1}\boldsymbol{\mu_u}, \tag{5}$$

$$\mathrm{Var}(f(\boldsymbol{x}^*)) = k(\boldsymbol{x}^*, \boldsymbol{x}^*) - k(\boldsymbol{x}^*, \boldsymbol{Z})k(\boldsymbol{Z}, \boldsymbol{Z})^{-1}\left(k(\boldsymbol{Z}, \boldsymbol{Z}) - \Sigma_{\boldsymbol{u}}\right)k(\boldsymbol{Z}, \boldsymbol{Z})^{-1}k(\boldsymbol{Z}, \boldsymbol{x}^*), \tag{6}$$

under the assumption of a constant zero prior mean function. This assumption is easily relaxed if needed. Here $\boldsymbol{Z}$ denotes the inducing *locations* in the input space, i.e. $f(\boldsymbol{Z}) = \boldsymbol{u} \sim \mathcal{N}(\boldsymbol{\mu_u}, \boldsymbol{\Sigma_u})$. Under further assumption of Gaussian observation noise $\epsilon$, i.e. $y^* = f(\boldsymbol{x}^*) + \epsilon$, then Titsias [2009] showed the optimal $\boldsymbol{\mu_u}$ and $\boldsymbol{\Sigma_u}$ are known analytically. In a *sought* analogy to Figure 1, $\boldsymbol{Z}$ would be the red points and $\boldsymbol{u}$ would be the orange.

## 2 Bézier Gaussian Processes

Inspired by Bézier surfaces, we construct a Gaussian process $f : [0, 1]^d \to \mathbb{R}$ as

$$f(\boldsymbol{x}) = \sum_{i_1=0}^{\nu_1} \sum_{i_2=0}^{\nu_2} \cdots \sum_{i_d=0}^{\nu_d} B_{i_1}^{\nu_1}(x_1) \cdots B_{i_d}^{\nu_d}(x_d) \boldsymbol{P}_{i_1,\ldots,i_d}, \tag{7}$$

where $\boldsymbol{P}_{i_1,i_2,\ldots,i_d} \sim \mathcal{N}(\boldsymbol{\vartheta}_{i_1,i_2,\ldots,i_d}, \boldsymbol{\Sigma}_{i_1,i_2,\ldots,i_d})$ are Gaussian variables and $\boldsymbol{x} = (x_1, x_2, \ldots, x_d)$. Here, $x_\gamma \in [0, 1]$ for $\gamma = 1, \ldots, d$. We write $\boldsymbol{P}$, with capital letter to emphasise that it is a random variable now. Further, we write it in boldface though we here only consider the scalar case, i.e. regression, but the multi-output case is not fundamentally different. It is easy to verify that $f$ satisfies the definition of a GP since it, for any $\boldsymbol{x}$, is a scaled sum of Gaussians. We assume that all $\boldsymbol{P}_{i_1,\ldots,i_d}$ are fully independent. With that assumption, we can make the following observation for the mean and kernel function

$$\mu(\boldsymbol{x}) := \sum_{i_1=0}^{\nu_1} \sum_{i_2=0}^{\nu_2} \cdots \sum_{i_d=0}^{\nu_d} B_{i_1}^{\nu_1}(x_1) \cdots B_{i_d}^{\nu_d}(x_d) \boldsymbol{\vartheta}_{i_1,\ldots,i_d}, \quad \text{and} \tag{8}$$

$$k(\boldsymbol{x}, \boldsymbol{z}) := \text{Cov}\left(f(\boldsymbol{x}), f(\boldsymbol{z})\right) = \sum_{i_1=0}^{\nu_1} \cdots \sum_{i_d=0}^{\nu_d} B_{i_1}^{\nu_1}(x_1) \cdots B_{i_d}^{\nu_d}(x_d) \boldsymbol{\Sigma}_{i_1,\ldots,i_d} B_{i_1}^{\nu_1}(z_1) \cdots B_{i_d}^{\nu_d}(z_d). \tag{9}$$

The Bernstein polynomials can approximate *any* continuous function given the order is large enough, thus they make a good basis for GP regression [Hildebrandt and Schoenberg, 1933]. Naturally, selecting a *prior* over $f$ comes down to selecting a prior over the random control points $\boldsymbol{P}$. The most common prior mean in GP regression, the constant zero function, is then easily obtained by $\boldsymbol{\vartheta}_{i_1,\ldots,i_d} = 0$ for all $\boldsymbol{i}$. By construction, the choice of $\boldsymbol{\Sigma}_{i_1,\ldots,i_d}$ needs consideration to yield a convenient prior over $f$. Mindlessly setting $\boldsymbol{\Sigma}_{i_1,\ldots,i_d} = 1$ would make $\text{Var}\left(f(\boldsymbol{x})\right)$ collapse to zero *quickly* in the central region of the domain, *especially* as dimensions $d$ grow. This, of course, gives a much too narrow prior over $f$.

Figure 2 (middle) shows that in the central region the standard deviation of $f$ is smaller due to the nature of the Bernstein polynomials. If we consider instead a two-dimensional input the standard deviation would collapse even more, as we would then see the shrinking effect for both dimension and multiply them. We can, however, adjust for this.

We define the *inverse squared Bernstein adjusted* prior to counter this effect. In all dimensions $\gamma = 1, \ldots, d$, let

$$\boldsymbol{\varsigma_\gamma} = \boldsymbol{A}_\gamma^{-1} \mathbf{1}_{\nu_\gamma+1}, \qquad \text{where } A_{i,j} = \left(B_j^{\nu_\gamma}(i/\nu_\gamma)\right)^2, \tag{10}$$

and $\nu_\gamma$ denotes the order of the dimension $\gamma$. Then setting $\boldsymbol{\Sigma}_{i_1,\ldots,i_d} = \prod_{\gamma=1}^d \varsigma_\gamma(i_\gamma)$ ensures that $\text{Var}\left(f(\boldsymbol{x})\right) \approx 1$ over the entire domain $[0, 1]^d$. Eq. 10 solves a linear system, such that $\text{Var}\left(f(i/\nu_\gamma)\right) = 1$, for $i = 0, \ldots, \nu_\gamma$. This means a prior hardly distinguishable from standard stationary ones such as the RBF kernel. Visual representation of this prior is shown in Figure 2 (right). This adjustment works up to $\nu_\gamma = 25$, after which negative values occur.

Summarising, we introduced a kernel based on Bézier surfaces. An alternative viewpoint is that $f$ is a polynomial GP, but with Bernstein basis rather than the canonical basis. We remark that $f$ is defined outside the domain $[0, 1]^d$; any intuition about the prior there is not considered, and we will not pursue investigating data points outside this domain. Of course, for practical purposes this domain generalises without loss of generality to *any* rectangular domain $[a_1, b_1] \times \ldots \times [a_d, b_d]$. For presentation purposes we keep writing $[0, 1]^d$. Next, we show how we infer an approximate posterior given data.

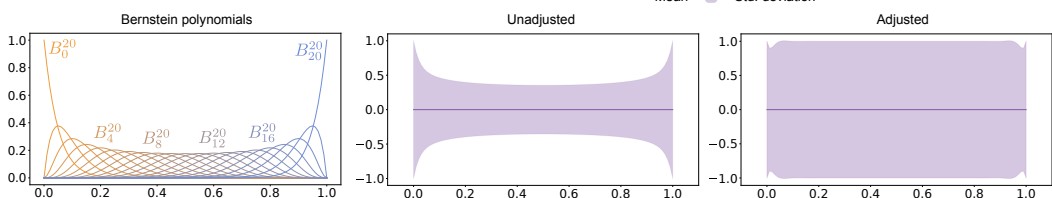

Figure 2: *Left*: The 21 Bernstein polynomials that make up the basis of a Bézier GP of order 20. We observe how they each impact a 'local' region along the [0,1] domain. *Middle*: If not accounting for the Bernstein polynomials behaviour, the variance is of $f$ is more narrow in the central region. *Right*: By adjusting, see Eq. 10, we can enforce a uniform variance over the domain.

## 2.1 Variational inference

Let $\boldsymbol{P}$ denote the set of all $\boldsymbol{P}_{i_1,\ldots,i_d}$. As the prior of $f$ is fully determined by the random control points $\boldsymbol{P}$, the posterior of $f$ is determined by the posterior of these. As per above, we set the prior

$$p(\boldsymbol{P}) = \prod_{i_1=0}^{\nu_1} \prod_{i_2=0}^{\nu_2} \cdots \prod_{i_d=0}^{\nu_d} p(\boldsymbol{P}_{i_1,\ldots,i_d}) := \prod_{i_1=0}^{\nu_1} \prod_{i_2=0}^{\nu_2} \cdots \prod_{i_d=0}^{\nu_d} \mathcal{N}(0, \boldsymbol{\Sigma}_{i_1,\ldots,i_d}), \tag{11}$$

where $\boldsymbol{\Sigma}_{i_1,\ldots,i_d} = \prod_{\gamma=1}^{d} \varsigma_\gamma(i_\gamma)$. We utilise variational inference to approximate the posterior of the control points, and hence $f$. This means we introduce *variational* control points. We assume they are fully independent (usually called the mean-field assumption), and have free parameters for the mean and variance, such that $\boldsymbol{P}_{i_1,\ldots,i_d} \sim \mathcal{N}(\hat{\boldsymbol{\vartheta}}_{i_1,i_2,\ldots,i_d}, \hat{\boldsymbol{\Sigma}}_{i_1,i_2,\ldots,i_d})$.

Assume we have observed data $\mathcal{D} = \{\boldsymbol{x}_j, y_j\}_{j=1}^{n}$. The key quantity in variational inference is the Kullback-Leibler divergence between the true posterior $p(\boldsymbol{P}|y)$ and the variational approximation – which we denote $q(\boldsymbol{P})$. The smaller divergence, the better approximation of the true posterior. Without access to the true posterior, the quantity is not computable. However, it has been shown this divergence is equal to the slack in Jensen's inequality used of the log-marginal likelihood: $\log p(y)$.

$$\log p(y) = \log \int p(y|\boldsymbol{P})p(\boldsymbol{P})\mathrm{d}\boldsymbol{P} \geq \int \log\left(\frac{p(y|\boldsymbol{P})p(\boldsymbol{P})}{q(\boldsymbol{P})}\right) q(\boldsymbol{P})\mathrm{d}\boldsymbol{P} \tag{12}$$

$$= \mathbb{E}_{q(\boldsymbol{P})}\left[\log p(y|\boldsymbol{P})\right] - \mathrm{KL}\left(q(\boldsymbol{P})\|p(\boldsymbol{P})\right). \tag{13}$$

Knowing this, we can approximate the true posterior with $q(\boldsymbol{P})$ by maximising Eq. (13). This is the evidence lower bound, and it is maximised with respect to the *variational parameters* $\hat{\boldsymbol{\vartheta}}_{i_1,i_2,\ldots,i_d}$ and $\hat{\boldsymbol{\Sigma}}_{i_1,i_2,\ldots,i_d}$. This is fully analytical when the variational parameters and a Gaussian likelihood is assumed.

We assume our observation model is disturbed with additive Gaussian noise, which in other words means our likelihood is Gaussian

$$p(y_j|\boldsymbol{P}) := \mathcal{N}\left(y_j|f(\boldsymbol{x}_j), \sigma^2\right), \quad \sigma^2 > 0, \tag{14}$$

for each $j = 1, \ldots, n$ and we assume they are independent conditioned on $\boldsymbol{P}$. With these assumption the first term in Eq. (13) becomes

$$\mathbb{E}_{q(\boldsymbol{P})}\left[\log p(y|\boldsymbol{P})\right] = -\frac{1}{2} \sum_{j=1}^{n} \log(2\pi) + \log(\sigma^2) + \frac{\left(y_j - \mathbb{E}_{q(\boldsymbol{P})}[f(\boldsymbol{x}_j)]\right)^2 + \mathrm{Var}_{q(\boldsymbol{P})}\left(f(\boldsymbol{x}_j)\right)}{\sigma^2}, \tag{15}$$

where $\mathbb{E}_{q(\boldsymbol{P})}[f(\boldsymbol{x}_j)]$ and $\mathrm{Var}_{q(\boldsymbol{P})}(f(\boldsymbol{x}_j))$ are given as Eq. (8) and (9) respectively, but with the variational parameters $\hat{\boldsymbol{\vartheta}}_{i_1,i_2,\ldots,i_d}$ and $\hat{\boldsymbol{\Sigma}}_{i_1,i_2,\ldots,i_d}$ used.

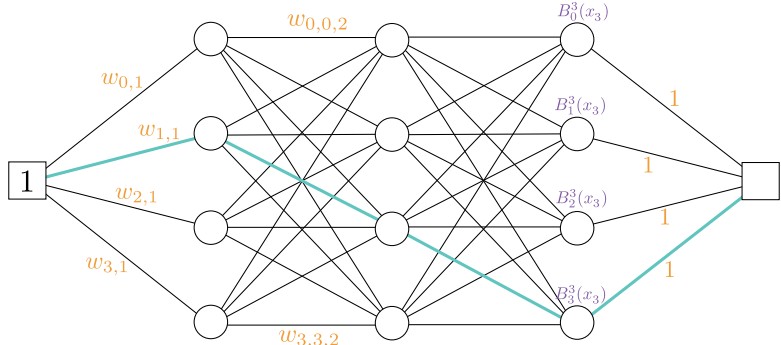

Figure 3: A Bézier buttress visualised. Here, the input dimension is 3, hence 3 layers. Each layer has 4 nodes, since each dimension has order 3. There exist $4^3$ paths from source (left square) to sink (right square), each path represents one unique control point. We can sum over all control points by sequential matrix multiplication from source to sink.

The second term in Eq. (13) enjoys the independence of control points to split into sums

$$\mathrm{KL}\left(q(\boldsymbol{P})\|p(\boldsymbol{P})\right) = \sum_{i_1=0}^{\nu_1}\sum_{i_2=0}^{\nu_2}\cdots\sum_{i_d=0}^{\nu_d}\mathrm{KL}\left(q(\boldsymbol{P}_{i_1,\ldots,i_d})\|p(\boldsymbol{P}_{i_1,\ldots,i_d})\right) \tag{16}$$

$$= \sum_{i_1=0}^{\nu_1}\sum_{i_2=0}^{\nu_2}\cdots\sum_{i_d=0}^{\nu_d}\left\{\frac{\hat{\boldsymbol{\Sigma}}_{i_1,i_2,\ldots,i_d}}{\boldsymbol{\Sigma}_{i_1,i_2,\ldots,i_d}} - 1 + \frac{\hat{\boldsymbol{\vartheta}}^2_{i_1,i_2,\ldots,i_d}}{\boldsymbol{\Sigma}_{i_1,i_2,\ldots,i_d}} + \log\frac{\boldsymbol{\Sigma}_{i_1,i_2,\ldots,i_d}}{\hat{\boldsymbol{\Sigma}}_{i_1,i_2,\ldots,i_d}}\right\}. \tag{17}$$

When inspecting the evidence lower bound, Eq. (13), we see it has a data term forcing control points to fit the data, and a KL-term to make control points revert to the prior. Knowing how control points are allocated in the input domain, we expect control points in regions of no data revert to the prior. This is similar to what stationary kernels do in said regions. We verify visually in Section 4.

All together, Bézier GPs can be adjusted to have priors similar to stationary GPs, and have analogous posterior behaviour, which is favourable to many practitioners. But Bézier GPs *scale*. None of the terms in the evidence lower bound require matrix inversions or determinants. It is simple to mini-batch over the data points, utilising stochastic variational inference [Hoffman et al., 2013], to scale it to large $n$. However, nearly all terms require evaluations of *huge* sums if the input dimension is high. The next section is aimed at this problem.

## 2.2 Scalability with the Bézier buttress

Until this point, we have omitted addressing the number of random control points needed for Bézier GPs. Let us denote this number $\tau$. It can quickly be checked that $\tau = \prod_{\gamma=1}^{d}(\nu_\gamma + 1)$. This implies that to evaluate $f$ we must sum over $\tau$ summands, which as $d$ increases, quickly becomes computationally cumbersome. $\tau$ increases exponentially with $d$. It is justifiable to view the random control points as inducing points; after all, they are Gaussian variables in the output space. Thus, it would be extremely valuable to manage exponentially many of them.

To overcome this, we introduce the *Bézier buttress*[2]. We assume parameters of the random control points, say $\boldsymbol{\vartheta}$, can parametrise $\boldsymbol{\vartheta}_{i_1,i_2,\ldots,i_d} = \prod_{\gamma=1}^{d} w_{i_{\gamma-1},i_\gamma,\gamma}$, where $w_{0,i_1,1} := w_{i_1,1}$. This assumption is the *key* of the Bézier buttress. Figure 3 provides visualisation. It visualises a source-sink graph, where each unique path from source to sink represents one unique control point with above parametrisation. The cyan highlighted path represents the $\boldsymbol{\vartheta}_{1,2,3} = w_{1,1}w_{1,2,2}w_{2,3,3}$, where we multiply the values along the path from source to sink. Notice last edges have value 1.

In the Bézier buttress there are $d$ layers, one for each input dimension, and $\nu_\gamma + 1$ nodes in each layer $\gamma = 1, \ldots, d$. Borrowing from neural network terminology, a forward-pass is a sequential series of matrix multiplications which are element-wise warped with non-linearities, such as $\tanh$ or ReLU. If

---

[2]A buttress is an architectural structure that provides support to a building.

we let our sequence of matrices be $\boldsymbol{w_1}, \boldsymbol{w_2}, \ldots, \boldsymbol{w_d}$, where $\boldsymbol{w_\gamma}$ is the matrix with entries $\{w_{i,k,\gamma}\}_{i,k}$. Let $\mathbf{1}_\nu$ denote the vector of size $\nu$ with 1 in all entries, then fixing 'the input' to $\mathbf{1}_{\nu_1+1}^\top$ and the last matrix to $\mathbf{1}_{\nu_d+1}$, a forward pass is

$$\mathbf{1}_{\nu_1+1}^\top \boldsymbol{w_1} \boldsymbol{w_2} \cdots \boldsymbol{w_d} \mathbf{1}_{\nu_d+1} = \sum_{i_1=0}^{\nu_1} \sum_{i_2=0}^{\nu_2} \cdots \sum_{i_d=0}^{\nu_d} \boldsymbol{\vartheta}_{i_1,i_2,\ldots,i_d}. \tag{18}$$

We see this forward pass computes a sum over all control points *means*. Naturally, we construct a Bézier buttress for summing over *variances* too, which must be restricted to positive weights. It was these sums that formed our bottleneck, in this parametrisation it is just a sequence of matrix products.

What about computing $f$? It comes down to a use of 'non-linearities' in the buttress. Multiplying *element-wise* the Bernstein polynomials as seen in Figure 3 (visualised only on 3rd layer), a forward pass computes either $\mathbb{E}[f(x)]$, or $\mathrm{Var}(f(x))$ if using squared Bernstein polynomials. Each control point is then exactly multiplied by the correct polynomials from its way from source to sink. Notice, the 'input' is fixed to 1 in the source, but the observed $\boldsymbol{x}$ is appearing via the Bernstein polynomials along the way. We can write this too as a sequence of matrix products

$$\mathbb{E}[f(\boldsymbol{x})] = \sum_{i_1=0}^{\nu_1} \sum_{i_2=0}^{\nu_2} \cdots \sum_{i_d=0}^{\nu_d} B_{i_1}^{\nu_1}(x_1) \cdots B_{i_d}^{\nu_d}(x_d) \boldsymbol{\vartheta}_{i_1,\ldots,i_d} = \mathbf{1}_{\nu_1+1}^\top \boldsymbol{w_1} \boldsymbol{B}_{x_1} \cdots \boldsymbol{w_d} \boldsymbol{B}_{x_d} \mathbf{1}_{\nu_d+1}, \tag{19}$$

where $\boldsymbol{B}_{x_\gamma}$ is a diagonal matrix with the $\nu_\gamma + 1$ Bernstein polynomials on its diagonal. For the variance of $f(\boldsymbol{x})$ there exists a similar expression, of course with squared polynomials, and with the positive weights associated with the variance Bézier buttress.

On this inspection, all terms needed to compute Eq. (15) are available. All terms for the KL divergences are algebraic manipulations of Eq. (18) – these are explicit in the supplementary material. The key takeaway for Bézier buttress is that we parametrise each random control point as a product of weights. This can be seen as an *amortisation*, and we do inference on these weights rather than the control points themselves. Hence, no matrix inversions are needed to backpropagate through our objective, and a forward pass is only a sequence of matrix products.

## 2.3 Marginalising matrix commutativity

Matrix multiplication is not commutative. This implies the ordering of the matrices in Eq. (18) matter, which again implies how we order the input dimensions in the Bézier buttress is of importance. This is the price for the computational benefit this parametrisation gives. An ordering of the input dimensions is somewhat unnatural for spatial regression, so we present a way to overcome this in approximate Bayesian manner. Define $f = f_1 + f_2 + \cdots + f_r$, where each of these individual $f_k$, $k \in \{1, \ldots, r\}$, are Bézier GPs with a random permutation of the ordering in the associated Bézier buttress. In other words, we let $f$ be an ensemble of Bézier GPs to (approximately) marginalise over all possible orderings. The number of all possible orderings quickly becomes too large for which to account; in practice, we set $r$ in a feasible region, say 20, which gives satisfactory empirical performance.

Another lens on this is that each control point is a sum of $r$ control points – each control point's standard deviation is scaled by $r^{-1}$, to obtain the same prior as so far discussed. Oddly, we have then circumvented the problem of too many control points by introducing even more control points. That is, each control point mean parametrise $\boldsymbol{\vartheta}_{i_1,i_2,\ldots,i_d} = \sum_{k=1}^{r} \prod_{\gamma=1}^{d} w_{i_{t_k(\gamma)-1}, i_{t_k(\gamma)}, t_k(\gamma)}$, where $t_k$ denotes a random permutation of $(1, \ldots, d)$. We remind again that similar expression exist for control point variances, restricted to positive weights.

As remarked, inference comes down to a forward pass in the Bézier buttress, a sequence of $d$ matrix multiplications. Assume all dimension are of order $\nu$, then the computational complexity of one forward pass is $\mathcal{O}\left(d(\nu+1)^2\right)$. Now we need $r$ forward passes to marginalise the ordering, and $n$ forward passes, one for each observation, leaving final complexity of $\mathcal{O}(nrd\nu^2)$. Linear in $n$ and $d$.

## 3 Related Work

Variational inference in GPs was initially considered by Csató et al. [1999] and Gibbs and MacKay [2000]. In recent times, the focus has shifted focus to scalable solutions to accommodate the big

data era. In this respect, Titsias [2009] took a variational approach to the inducing points methods [Quinonero-Candela and Rasmussen, 2005]; later Hensman et al. [2013] further enhanced scalability to allow for mini-batching and a wider class of likelihoods. Still, the need for more inducing points is of importance, especially as the number of input features grows.

The response to this has mostly revolved around exploiting some structure of the kernel. Wilson and Nickisch [2015] and Wu et al. [2021] exploit specific structure that allow for fast linear algebra methods; similar to our method inducing locations tend to lie on grid. These grids expand fast as the input dimension increases, as also pointed out earlier in the article. Kapoor et al. [2021] remedy this by instead of a rectangular grid, they consider the permutohedral lattice, such that each observation only embeds to $d + 1$ neighbours instead of $2^d$, as in [Wilson and Nickisch, 2015].

Another approach to allowing for more inducing points is incorporating nearest neighbour search in the approximation [Kim et al., 2005, Nguyen-Tuong et al., 2008]. Tran et al. [2021] introduced sparse-within-sparse where they have many inducing points in memory, but each time search for the $k$-nearest ones to any observation. They discard the remaining ones as they have little to no influence on the observation. Wu et al. [2022] made a variational pendant to this method.

Lastly, when dealing with high-dimensional inputs it is worth mentioning *feature learning*. That is, learning more low-dimensional features where GPs have better performance. The success of deep models has been transported to GPs by Damianou and Lawrence [2013] and later scaled to large datasets in [Salimbeni and Deisenroth, 2017]. Another approach is Deep Kernel Learning [Wilson et al., 2016, Bradshaw et al., 2017], where feature extraction happens inside the kernel function; lately Ober et al. [2021] has investigated the limitations and benefits of these models.

We have treated structured control points as our version of inducing points; and by parametrising them with a Bézier buttress, we limit the expansion of grids to linear growth in parameters. We are not the first to consider the Bernstein polynomials as a basis for learning functions. Petrone [1999b] used it to model kernel estimate probability density functions, and several follow up works [Petrone, 1999a, Petrone and Wasserman, 2002]. Hug et al. [2020] recently introduced Bézier GPs, but with a focus on time series [Hug et al., 2022]. Our emphasis has been on spatial input; even the Bézier surface literature contain close to nothing on more than 2-dimensional surfaces.

## 4 Evaluation

We split our evaluation into four parts. First, we visually inspect the posterior on a one dimensional toy dataset to show how the control points behave, and indicate that there indeed is a stationary-like behaviour on the domain of the hypercube. Next, we test empirically on some standard UCI Benchmark datasets, to gives insight into when Bézier GPs are applicable. After that, we switch to tall and wide data – large both in the input dimension and in number of data points. These experiments give certainty that the method delivers on its key promise: scalability. Lastly, we turn our eyes to the method itself and investigate how performance is influenced by the ordering of dimensions.

Care is needed in optimising a Bézier GP – not all parameters are born equal. We split optimisation into two phases. First, we optimise all variational parameters, keeping the likelihood variance $\sigma^2$ fixed as $\tau^{-1}$, with $\tau$ being the number of control points. After this initial phase, we optimise $\sigma^2$ with all variational parameters fixed. We let both phases run for 10000 iterations with a mini-batch size of 500, for all datasets. Both phases use the Adam optimiser [Kingma and Ba, 2015], the first phase with learning rate 0.001, and the second with learning rate 0.01. If not following a such a bespoke training scheme, we see a tendency for the posterior to revert to the prior, because the KL-term becomes too dominating initially. This training scheme is designed for the Gaussian likelihood, but we wish to emphasise that, in principle, the loss function is accurate for any choice of likelihood.

### 4.1 One dimensional visual inspection

We hypothesised the objective function, Eq. 16, would ensure, within the hypercube domain, that $f$ reverts to its prior in regions where data is scarce. To verify this we construct a small dataset to inspect. We generate one-dimensional inputs uniformly in the regions $[0, 0.33]$ and $[0.66, 1]$; we sample 20 observation in each region. The responsive variable is generated as $y(x) = 3 \sin(16x)$. According to the hypothesis, $f$ should in the region $[0.33, 0.66]$, tend towards zero in mean, and increase its variation here. We use a BézierGP of order 20 to model the observations, since they are highly

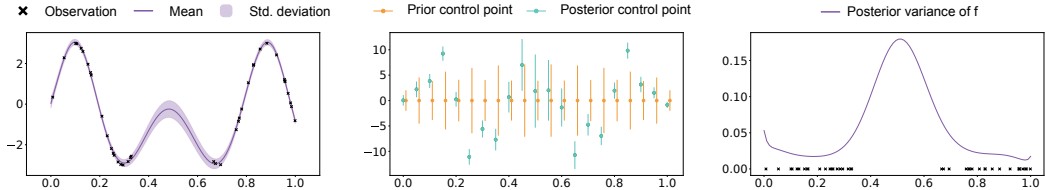

Figure 4: *Left*: Posterior distribution of $f$. *Middle*: Posterior and prior distribution of control points. *Right*: Posterior variance as function over the domain. Variance increases in scarce data regions.

| | | power | protein | energy | boston | bike | keggdirected | concrete | elevators |
|---|---|---|---|---|---|---|---|---|---|
| | $n$ | 9568 | 45730 | 768 | 506 | 17379 | 48837 | 1030 | 16599 |
| | $d$ | 4 | 9 | 8 | 13 | 17 | 20 | 8 | 18 |
| | | | | | Test log-likelihood | | | | |
| SGPR | $m = 100$ | $-2.7789 \pm 0.04$ | $-2.9307 \pm 0.01$ | $-1.6450 \pm 0.07$ | $-2.4993 \pm 0.27$ | $-0.6661 \pm 0.03$ | $0.6498 \pm 0.03$ | $-3.1839 \pm 0.07$ | $0.9167 \pm 0.02$ |
| | $m = 500$ | $-2.7440 \pm 0.04$ | $-2.8479 \pm 0.04$ | $-0.7707 \pm 0.13$ | $-2.4592 \pm 0.33$ | $-0.4680 \pm 0.04$ | $0.7133 \pm 0.03$ | $-3.0658 \pm 0.08$ | $0.9373 \pm 0.02$ |
| SimplexGP | | $-3.4416 \pm 0.06$ | $-3.2745 \pm 0.04$ | NA | NA | $-1.0932 \pm 0.19$ | $-2.3241 \pm 5.13$ | $-4.0338 \pm 0.02$ | $-0.2633 \pm 0.01$ |
| BezierGP | $\nu = 5$ | $-2.8015 \pm 0.04$ | $-2.9585 \pm 0.01$ | $-0.6197 \pm 0.25$ | $-45.479 \pm 18.6$ | $0.4724 \pm 0.14$ | $0.6589 \pm 0.06$ | $-3.4612 \pm 0.62$ | $0.9534 \pm 0.02$ |
| | $\nu = 10$ | $-2.7736 \pm 0.04$ | $-2.9257 \pm 0.01$ | $-0.7163 \pm 0.46$ | $-197.23 \pm 140.5$ | $0.7020 \pm 0.18$ | $0.6789 \pm 0.06$ | $-4.5613 \pm 1.20$ | $0.9565 \pm 0.02$ |
| | $\nu = 20$ | $-2.7391 \pm 0.05$ | $-2.8902 \pm 0.01$ | $-0.6504 \pm 0.62$ | $-139.42 \pm 52.7$ | $0.7475 \pm 0.32$ | $0.6939 \pm 0.06$ | $-7.7174 \pm 2.70$ | $0.9058 \pm 0.03$ |
| | | | | | Test RMSE | | | | |
| SGPR | $m = 100$ | $3.8806 \pm 0.15$ | $4.5272 \pm 0.04$ | $1.1562 \pm 0.11$ | $2.9372 \pm 0.65$ | $0.4665 \pm 0.02$ | $0.1279 \pm 0.00$ | $5.8980 \pm 0.54$ | $0.0968 \pm 0.00$ |
| | $m = 500$ | $3.7383 \pm 0.16$ | $4.1755 \pm 0.16$ | $0.5664 \pm 0.12$ | $2.8233 \pm 0.65$ | $0.3829 \pm 0.02$ | $0.1212 \pm 0.01$ | $5.3522 \pm 0.80$ | $0.0948 \pm 0.00$ |
| SimplexGP | | $3.1147 \pm 0.26$ | $4.1271 \pm 0.13$ | NA | NA | $0.2876 \pm 0.07$ | $2.6439 \pm 2.29$ | $5.4457 \pm 0.75$ | $0.1256 \pm 0.01$ |
| BezierGP | $\nu = 5$ | $3.9750 \pm 0.15$ | $4.6620 \pm 0.04$ | $0.4348 \pm 0.08$ | $4.7007 \pm 0.91$ | $0.1474 \pm 0.02$ | $0.1319 \pm 0.03$ | $4.8127 \pm 0.86$ | $0.2939 \pm 0.85$ |
| | $\nu = 10$ | $3.8675 \pm 0.15$ | $4.5112 \pm 0.04$ | $0.4157 \pm 0.11$ | $5.6712 \pm 1.76$ | $0.1100 \pm 0.01$ | $0.1735 \pm 0.22$ | $4.7917 \pm 0.94$ | $1023.5 \pm 4e3$ |
| | $\nu = 20$ | $3.7427 \pm 0.18$ | $4.3538 \pm 0.04$ | $0.3573 \pm 0.11$ | $4.9404 \pm 2.93$ | $0.0821 \pm 0.01$ | $80.05 \pm 348.4$ | $4.9829 \pm 0.95$ | $7e11 \pm 3e12$ |

Table 1: Results on eight standard UCI Benchmark datasets. We list test-set log-likehood and RMSE, both are averages over 20 train/test splits. On top are test log-likelihoods (higher is better), and bottom is test RMSE (lower is better). NA indicates Cholesky error.

non-linear. Figure 4 shows the posterior distribution of $f$ to the left; we observe $f$ tends towards the prior in the middle region. The middle plot illustrates the distribution, both prior and posterior, of the 21 control points. There is a clear tendency for the central-most points to align the posterior and posterior, enforcing this behaviour in $f$. The non-equal priors are due to the inverse-squared Bernstein adjusted prior which ensures a uniform variation in $f$ over the domain, see Figure 2. The plot to the right in Figure 4 shows the behaviour foundational to practitioners of Bayesian optimisation and active learning etc., the variance increase away from data regions.

## 4.2 UCI Benchmark

We evaluate on eight small to mid-size real world datasets commonly used to benchmark regression [Hernandez-Lobato and Adams, 2015]. We split each dataset into train/test-split with the ratio $90/10$. We do this over 20 random splits and report test set RMSE and log-likelihood average and standard deviation over splits. We choose baselines to be SGPR, following the method from Titsias [2009]; we do both for 100 and 500 inducing variables. *SimplexGP* is another baseline, they suggest their approximation is beneficial for semi-high dimensional inputs (between 3 and 20) [Kapoor et al., 2021], hence they are an obvious baseline. SimplexGP usually use a validation set to choose the final model. This is due to a highly noisy optimisation scheme using a high error-tolerance (1.0) for conjugate gradients. We remedy this by setting the error-tolerance to (0.01), which harms scalability, but we can omit using a validation set for better comparability. Wang et al. [2019] recommend this error-tolerance, but remark it is more stable for RMSE than for log-likelihood. For BézierGP, we fix the number of permutations to $r = 20$, and vary the order in $\nu = 5, 10, 20$. The order is identical over input dimensions. The inputs are pre-processed such that the training set is contained in $[0, 1]^d$.

Table 1 contains the results of this experiment. We make the following observations about our presented BézierGP. On *keggdirected* and *elevators* there are test points outside the defined domain on some splits, which cause the RMSE to be extreme, but the likelihood is more forgiving. This highlights the constraint of our model: it needs a box-bounded domain to be a priori known. Had we standardised such that *both* test and train data were in the hypercube, BézierGP ($\nu = 20$) would have a average test RMSE of 0.0937. We could not reproduce results from Kapoor et al. [2021] on *keggdirected*, the optimisation was too noisy and with no use of validation. On *concrete, boston* and *energy* we see overfitting tendencies. Even though BézierGP is the optimal choice on *energy*

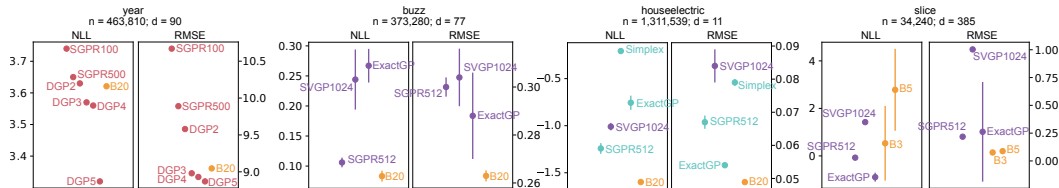

Figure 5: Test *negative* log-likelihood (lower is better) and RMSE (lower is better) for large scale regression. Colours indicate the origin of numbers: red from Salimbeni and Deisenroth [2017], purple from Wang et al. [2019], and cyan from Kapoor et al. [2021]. Orange is ours. We observe our BézierGP is highly competitive on large datasets.

there is a mismatch between train and test error. On *concrete* this shows in better test RMSE, than the baselines, but the variance is overfitted yielding non-optimal likelihood. We conjecture this happens because the $n/d$-ratio is low; which makes it more likely to overfit the control points – especially for higher orders $\nu$. Knowing these model fallacies, we observe that BézierGP outperforms on baselines on multiple datasets, most notably the 17-dimensional *bike* dataset.

### 4.3 Large scale regression

Figure 5 shows the results of regression tasks in regimes of high dimensions and one in high number of observations. Here, we follow exactly the experimental setup of either Salimbeni and Deisenroth [2017] or Wang et al. [2019]. If the latter, we use the validation-split they use as training data. Our optimisation scheme for BézierGP is consistent with above, except for *slice*, where the first training phase runs for 30000 iterations. We discard test points that are not in the $[0, 1]^d$ domain – in no situation did this remove more than $0.001\%$ of the test set. The number after DGP, denotes the number of hidden layers in a Deep GP [Salimbeni and Deisenroth, 2017], after SGPR and SVGP it denotes the number of inducing points. SVGP refers to the method from Hensman et al. [2013]. After B, it denotes the order used in BézierGP.

On *year*, we observe our (non-deep) model is on-par with 2-layered Deep GPs, and closer to 3 in RMSE. The highest dimensional dataset, *slice*, sees us in the low $n/d$-ratio again, and we are again faced with a too flexible model. This is why we report results for orders 3 and 5, rather than 20, since the overfitting kicks in. Even for these small orders the test log-likelihood has high variance and under-performs compared to RMSE. With respect to RMSE it is top-performer signalling again it is overfitting the variance. On the remaining two datasets BézierGP is best-performing among baselines.

|  | power (24) | | protein (362880) | | bike (3e14) | |
|---|---|---|---|---|---|---|
|  | RMSE | LL | RMSE | LL | RMSE | LL |
| $r = 1$ | 4.1476 | −2.8418 | 4.8416 | −2.9962 | 0.1477 | 0.4813 |
| $r = 10$ | 3.8028 | −2.7549 | 4.4306 | −2.9080 | 0.0926 | 0.7275 |
| $r = 20$ | 4.2514 | −2.8826 | 4.3713 | −2.8943 | 0.0853 | 0.5454 |
| $r = 30$ | 3.8257 | −2.7551 | 4.3377 | −2.8867 | 0.0775 | 0.8372 |
| $r = 40$ | 3.6612 | −2.7190 | 4.3449 | −2.8886 | 0.0848 | 0.5024 |
| $r = 50$ | 3.7841 | −2.7503 | 4.2614 | −2.8686 | 0.0718 | 0.9009 |

Table 2: Performance in test RMSE and log-likelihood against the number of permutations, $r$, on three datasets. In parenthesis shows the number of possible permutations of input dimensions. Higher dimensional datasets show improving performance with increasing $r$.

### 4.4 Influence of number of permutations

All experiments so far used $r = 20$; that is, 20 random permutations of the ordering of dimension used in the Bézier buttress. For a problem with input dimension $d$, there exist $d!$ possible permutations. Table 2 shows results with varying $r$; for each dataset, the results are over the same train/test split (0.9/0.1). We fixed $\nu = 20$. Up to some noise in the optimisation phase, we see for the two highest dimensional datasets, *protein* and *bike*, performance improves with higher $r$. *Bike* has over $50\%$ reduction in RMSE from $r = 1$ to $r = 50$.

Table 2 emphasises the results we have presented are not optimised over hyperparameter $r$ and $\nu$. They also illustrate an interesting direction of future research: optimising these hyperparameters. We chose permutations by random sampling, but choosing them in a principled deliberate manner could yield good performance with a computationally manageable $r$. This result indicates, at least, *protein* and *bike* would see increased performance in Table 1 from better (or just more) permutations.

# 5 Discussion

We introduced the Bézier Gaussian Process – a GP, with a polynomial kernel in the Bernstein basis, that scales to a large number of observations, and remains space-filling for high number of input features (limited to a box-bounded domain). We illustrated that, with slight adjustments, the prior and posterior have similar behaviour to 'usual' stationary kernels. We presented the Bézier buttress, a weighted graph to which we amortise the inference, rather than inferring the control points themselves. The Bézier buttress allows GP inference without any matrix inversion. Using the Bézier buttress, we inferred $6^{385}$ control points for the high-dimensional *slice* dataset.

We highlighted weaknesses of the proposed model: most crucially the tendency of overfitting when the $n/d$-ratio is low. The results demonstrate scalability in both $n$ and $d$, but does not solve the *short, but wide* problem. The paper did not optimise over the hyperparameters of the proposed kernel, namely $\nu$ and $r$, but it showcased briefly that doing so might enhance BézierGPs empirically; especially smart selection of the permutations is an interesting direction for future research. We speculate that optimising over orders, on a validation set, would alleviate some of the overfitting issues.

## Acknowledgments and Disclosure of Funding

MJ is supported by the Carlsberg Foundation.

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
