# – *Appendix* –
# Bézier Gaussian Processes for Tall and Wide Data

**Martin Jørgensen**
Department of Engineering Science
University of Oxford
martinj@robots.ox.ac.uk

**Michael A. Osborne**
Department of Engineering Science
University of Oxford
mosb@robots.ox.ac.uk

## 1  Computations in the Bézier Buttress

This sections seeks to explain how the KL-divergence is computed using the Bézier buttress. It further explains more detailed parametrisation in the architecture. For completeness we here give a forward pass to compute $\mathrm{Var}\left(f(\boldsymbol{x})\right)$.

$$\mathrm{Var}\left(f(\boldsymbol{x})\right) = \mathbf{1}_{\nu_1+1}^{\top} \boldsymbol{w}_1 \boldsymbol{B}_{x_1}^2 \cdots \boldsymbol{w}_d \boldsymbol{B}_{x_d}^2 \mathbf{1}_{\nu_d+1}, \tag{1}$$

here we make the choice that $\{\boldsymbol{w}_\gamma\}_{i,j} := \exp(v_{i,j})\varsigma_{\gamma_i}$. This ensures positive weights and hence a positive output for the variance of $f$. $\varsigma_\gamma$ comes from the inverse squared Bernstein adjusted prior (see Section 2). $v$ are free parameters to be inferred in the variational posterior. This parametrisation makes computing the KL terms easier.

We remark all the following calculation are only for *one* Bézier buttress. Are there multiple Bézier buttresses, with different orderings of layers, the computations are equivalent for all of them.

For computing the KL we first recall from the paper

$$\mathrm{KL}\left(q(\boldsymbol{P})\|p(\boldsymbol{P})\right) = \sum_{i_1=0}^{\nu_1}\sum_{i_2=0}^{\nu_2}\cdots\sum_{i_d=0}^{\nu_d} \mathrm{KL}\left(q(\boldsymbol{P}_{i_1,\dots,i_d})\|p(\boldsymbol{P}_{i_1,\dots,i_d})\right) \tag{2}$$

$$= \sum_{i_1=0}^{\nu_1}\sum_{i_2=0}^{\nu_2}\cdots\sum_{i_d=0}^{\nu_d} \left\{ \frac{\hat{\boldsymbol{\Sigma}}_{i_1,i_2,\dots,i_d}}{\boldsymbol{\Sigma}_{i_1,i_2,\dots,i_d}} - 1 + \frac{\hat{\boldsymbol{\vartheta}}_{i_1,i_2,\dots,i_d}^2}{\boldsymbol{\Sigma}_{i_1,i_2,\dots,i_d}} + \log\frac{\boldsymbol{\Sigma}_{i_1,i_2,\dots,i_d}}{\hat{\boldsymbol{\Sigma}}_{i_1,i_2,\dots,i_d}} \right\}. \tag{3}$$

For easier reference we declare

$$S_1 := \sum_{i_1=0}^{\nu_1}\sum_{i_2=0}^{\nu_2}\cdots\sum_{i_d=0}^{\nu_d} \frac{\hat{\boldsymbol{\Sigma}}_{i_1,i_2,\dots,i_d}}{\boldsymbol{\Sigma}_{i_1,i_2,\dots,i_d}}, \tag{4}$$

$$S_2 := \sum_{i_1=0}^{\nu_1}\sum_{i_2=0}^{\nu_2}\cdots\sum_{i_d=0}^{\nu_d} \frac{\hat{\boldsymbol{\vartheta}}_{i_1,i_2,\dots,i_d}^2}{\boldsymbol{\Sigma}_{i_1,i_2,\dots,i_d}}, \tag{5}$$

$$S_3 := \sum_{i_1=0}^{\nu_1}\sum_{i_2=0}^{\nu_2}\cdots\sum_{i_d=0}^{\nu_d} \log\frac{\boldsymbol{\Sigma}_{i_1,i_2,\dots,i_d}}{\hat{\boldsymbol{\Sigma}}_{i_1,i_2,\dots,i_d}}. \tag{6}$$

We remind again that "hat" notation refers to parameters from variational posterior $q$. We also remind that the prior variance is given $\boldsymbol{\Sigma}_{i_1,\dots,i_d} = \prod_{\gamma=1}^{d} \varsigma_\gamma(i_\gamma)$. Because we have included $\varsigma$ in the parametrisation in the posterior $\hat{\boldsymbol{\Sigma}}_{i_1,\dots,i_d}$, they are cancelling out in the expression in $S_1$ and $S_3$. We get

$$S_1 = \mathbf{1}_{\nu_1+1}^{\top} \exp\boldsymbol{v}_1 \cdots \exp\boldsymbol{v}_d \mathbf{1}_{\nu_d+1}. \tag{7}$$

where $\exp$ is element-wise on the matrices.

36th Conference on Neural Information Processing Systems (NeurIPS 2022).

For $S_3$ we make the observation, based again on $\varsigma$ cancelling out in the fraction, that

$$\log \frac{\boldsymbol{\Sigma}_{i_1,i_2,\ldots,i_d}}{\hat{\boldsymbol{\Sigma}}_{i_1,i_2,\ldots,i_d}} = -\log \prod_{\gamma=1}^{d} \exp v_{i_\gamma-1,i_\gamma,\gamma} = \sum_{\gamma=1}^{d} v_{i_\gamma-1,i_\gamma,\gamma}. \tag{8}$$

That is, summing over $\log \hat{\boldsymbol{\Sigma}}_{i_1,i_2,\ldots,i_d}$ is basically counting how many paths (i.e. control points) use $v_{i_\gamma-1,i_\gamma,\gamma}$. That is determined as $\psi_\gamma = \frac{\tau}{(\nu_{\gamma-1}+1)(\nu_\gamma+1)}$. Here $\nu_0 := 0$. Hence,

$$-S_3 = \sum_{\gamma=1}^{d} \psi_\gamma \bigoplus \boldsymbol{v}_\gamma, \tag{9}$$

where $\bigoplus$ denotes summing the elements in the matrix.

Notice how the *variational* parametrisation of $\{\boldsymbol{w}_\gamma\}_{i,j} := \exp(v_{i,j})\varsigma_{\gamma_i}$ was carefully chosen for easily computing $S_1$ and $S_3$.

$S_2$ is more close to what described in main paper. We simply just need to square all the weights and correct with the prior variance. That is, correct with $1/\varsigma_\gamma$. Hence,

$$S_2 = \mathbf{1}_{\nu_1+1}^{\top} \boldsymbol{w}_1^2 \varsigma_1^{-1} \cdots \boldsymbol{w}_d^2 \varsigma_d^{-1} \mathbf{1}_{\nu_d+1}, \tag{10}$$

where here $\varsigma_\gamma$ is the diagonal matrix with $\varsigma_{\gamma_i}$ along its diagonal, for $i = 1, \ldots, \nu_\gamma$. Notice further here $\boldsymbol{w}$ are the weights in the *mean* Bézier buttress.

Now

$$\mathrm{KL}\left(q(\boldsymbol{P})\|p(\boldsymbol{P})\right) = S_1 - \tau + S_2 + S_3, \tag{11}$$

all of which are computed in a single forward pass in the Bézier buttress. $\tau$ is the number of all control points (in one buttress).

## 2  Numerical results

For reproducibility we give the values used to generate Figure 4. These are given in Table 1.

| year | buzz | houseelectric | slice | |
|---|---|---|---|---|
| | | Test log-likelihood | | |
| B20: $-3.6209 \pm 0.00$ | B20: $-0.0832 \pm 0.01$ | B20: $1.5987 \pm 0.00$ | B3: $-0.5321 \pm 1.36$ | B5: $-2.7831 \pm 1.49$ |
| | | Test RMSE | | |
| B20: $9.0461 \pm 0.01$ | B20: $0.2629 \pm 0.00$ | B20: $0.0489 \pm 0.00$ | B3: $0.0761 \pm 0.01$ | B5: $0.0880 \pm 0.02$ |

Table 1: Numerical values used create Figure 4. Here is listed average and standard deviation over 3 splits. On *year* the test-set was not standardised to compare with baselines there.