# OpenReview forum: "Bezier Gaussian Processes for Tall and Wide Data"
_NeurIPS.cc/2022/Conference — NeurIPS 2022 Accept_

### Official Review · Reviewer_qTRQ · 2022-07-09

**Rating:** 6
**Confidence:** 4
**Soundness:** 3 good
**Presentation:** 3 good
**Contribution:** 3 good

**Summary:**

The paper proposes a new Bézier buttress-based kernel that allows for scaling GP regression both in number of inputs (tall data) and in number of input features (wide data). The scalability is shown both theoretically and empirically through evaluation on popular datasets.


**Questions:**

I have a several questions to the authors.

1. Am I correct in understanding that computing full covariance would be just rendered as another type of "non-linearity" in the Bézier buttress, that is, multiplying be a cross-product of Bernstein polynomials instead of the squares of them?
2. Have you tried to make even larger orders $\nu$ work? Does it make sense to try larger orders of the Bézier?
3. Have you tried to faithfully compute the true posterior (on a small example perhaps) and see how good the proposed variational approximation is?
4. How easy is it to introduce non-Gaussian likelihood?
5. In your opinion, does it make sense to vary the order of each dimension (that is, non-equal $\nu_\gamma$)? It would make sense as not all dimensions are equal. That would just render a Bézier buttress with different layer widths.
6. A major thing is an assumption of the line 142 which makes Bézier buttress possible. This introduces inter-relations between variational parameters. Do you think this is not very restrictive of an assumption? Again, comparing to the true posterior would probably be good enough of an answer.
7. What happens when jointly optimizing likelihood noise and variational parameters? Have you observed a worsening of the performance?


**Limitations:**

The authors address the limitations and constraint of their proposed method. The distribution of the data must be supported on an a-priori known compact cube. Another constraint is a tendency to overfit when $n/d$ ratio is small. Orders $\nu$ larger than 25 do not work.



**Strengths And Weaknesses:**

*Originality*. The work is original. While there are other papers leveraging Bézier curves for Gaussian processes, the paper provides a way to parametrize Bézier control points to allow for a scalable training and inference (termed Bézier buttress).

*Quality*. The research is sound. The statements the authors make are supported by evidence. The authors make a detailed analysis of empirical results, addressing both advantages and limitations of the suggested method. They provide a review of the related work, and highlight areas of possible further research.

That said, a more detailed overview of performance against order of the Bézier would be nice.
Another thing beneficial for the paper would be reports on the wall time consumed and computing power used in the experiments.

*Clarity*. The work is written in a clear language, the presentation is on-point and laid out in a way that facilitates reading. The notation might be cumbersome sometimes.

One thing to pay attention to is in the supplementary material, line 5: the definition of the variance buttress weights. I think there is a forgotten dependence of $v$ on $\gamma$ and also $\zeta_{\gamma_i}$ doesn't make sense, I guess it should be $\zeta_\gamma(i_\gamma)$.
There is also a typo on line 142: "can be parametrized".

*Significance*. Since applying Gaussian processes to high-dimensional inputs is known to be hard, the proposed method that deals with both "wide" (high-dimensional inputs) and "tall" (many inputs) data is a significant contribution to the world of Gaussian process machine learning.

---

> ### Author Response · Authors · 2022-08-01
> **Reviewer qTRQ**
>
> We appreciate the time taken to review our paper and the insights provided with your discussion. We’re excited about the positive assessment and hope to clarify issues in our response.
>
> * Wall time and computing power
>
> With purpose, we did not list wall-time spent. This is because we did not use any stopping rule in the optimisation, hence wall-time should not be compared. All BezierGP experiments were done a (AMD Ryzen) 8-core laptop, except one (slice dataset) which was on a CPU with high memory. It was implemented using GPflow, based on Tensorflow, as GPU acceleration is available. To give some intuition on the time, training a BezierGP to convergence on a dataset like "elevators" takes at most 2-3 minutes on a laptop.
>
> * Line 5 Appendix
>
> Thank you for pointing this out. We will correct this.
>
> * Full covariance
>
> Yes, it is correctly understood that you can compute the covariance between two points by using the product of the corresponding Bernstein polynomials as the "non-linearity" in the Bezier buttress. We hope this is what you mean by "full" covariance. If not, please let us know.
>
> * Even larger orders
>
> We only tried up to 25, as when going larger our prior adjustment was not permissible. It would be interesting if there were other adjustments that would allow us to both have a sensible prior and go to very large orders.
>
> * True posterior
>
> You are right that comparing to the true posterior would be the optimal scenario. We tried to give some of this intuition in the simple case of a 1-d visualisation in Figure 4. Unfortunately, we cannot compute exact posteriors in this model. We are, as so many others, bound to use the mean-field assumption and variational inference. So for now our sanity checks of the posteriors have been inspected, visually and testing at odd locations (away from data). The variational approximations have passed our inspections, that is, it satisfies the desiderata that we had to the posterior. That is extrapolates high variance away from data, and has a sensible variance estimate within data regions.
>
> * Non-Gaussian likelihoods
>
> It should be mathematically straightforward to introduce non-Gaussian likelihoods. One issue that would arise would be to find an optimisation routine that accounts for large KL's in the initial phase (we fix the likelihood variance small in the initial phase). Yet, as also pointed out by reviewer W66W, this is effectively a KL annealing scheme, so there is reason to believe it should work for other likelihoods too. Mathematically, nothing hinders any other choice of likelihood.
>
> * Different orders per dimension
>
> We believe this one to potentially have a big impact. This is particularly in the case where some features are binary (or other finite discrete features). Here we could set the order of that feature to be 1 (effectively giving that feature 2 control points to vary), which would make it exact without unnecessary extra computation. This makes BezierGP also very applicable for inputs that are not Euclidean in nature.
> Also, we hope future work will try to optimise over orders for each dimension potentially finding features that are, say, linear. This should, hopefully, improve upon generalisation.
> And yes, it would ultimately be as easy as a Bezier buttress with varying widths in the layers.
>
> * Line 142 assumption
>
> Yes, this assumption is restrictive! What we can say at this time is that for 1 and 2 dimensional inputs the current parametrisation has the same "capacity" as free parametrisation. By free parametrisation we mean one parameter per control point, ie. no factorisation structure. For 3 dimensional input if we include all orderings the same holds. We are working on a deeper and more general framework to proof that for any dimension the factorisation structure have same capacity as the free parametrisation if we include all orderings of the dimensions. This will hopefully give more insights into how many permutations (orderings) are needed for good approximation of the free parametrisation.
>
> * Joint optimisation
>
> As mentioned above, what we observed with joint optimisation was that the KL-term would get all attention from the optimiser. Effectively, it would then just focus on minimising KL, and hence just "learn" the prior. So yes, the wrong minimum would be found and performance would be bad since it predicts zero everywhere. However, a quite simple KL annealing (see above) goes around this pathological optimum.

---

> > ### Comment · Reviewer_qTRQ · 2022-08-08
> > **Thanks**
> >
> > Thank you for your answers on the raised questions! I think the authors will agree there is a lot of room for future developments/improvement. My score remains unchanged; overall, I am positive about the paper.

---

### Official Review · Reviewer_mrad · 2022-07-12

**Rating:** 6
**Confidence:** 4
**Soundness:** 3 good
**Presentation:** 2 fair
**Contribution:** 3 good

**Summary:**

This work introduces a type of kernel for Gaussian process regression that requires only a linear computational cost in both number of observations and input features. The scaling is achieved by means of the introduction of the Bézier buttress structure, which allows approximate inference without computing matrix inverses or determinants.

**Questions:**

---Specific comments---

lines 13 to 16:
It might be useful to include references for Gaussian process model and the approximate methods for their scalability.

line 33:
Include references for Bayesian optimisation, active learning and reinforcement learning.

line 63:
As per Titsias' work (2009), I would suggest to use inducing variables when referring to $\boldsymbol{u}$, and inducing inputs/locations or points when referring to $\mathbf{Z}$.

line 65:
bold for $\boldsymbol{\Sigma_u}$?

line 72:
Why was the lower case variable $\mathbf{p}_{i_1,...,i_d}$  from Eq. (4)

changed to upper case variable $\mathbf{P}_{i_1,...,i_d}$?

is it a matrix now? What is the dimensionality of $\mathbf{P}_{i_1,...,i_d}$? Is it the same of Equation (1) where it was defined $\mathbf{p}_i \in \mathbb{R}^D$?
Is it just a scalar? if so, then most of the bold notations are problematic.

line 73:
It is necessary to introduce the mean $\boldsymbol{\vartheta}_{i_1,...,i_d}$

and covariance $\boldsymbol{\Sigma}_{i_1,...,i_d}$ of the Gaussian variables.

Since they are defined in bold it means they are not scalars, doesn't it?

line 81:
Typo? shouldn't it be $\mathbf{0}$ since the mean is a vector?
Also why bold "$i$" in "for all $\mathbf{i}$"? What is $\mathbf{i}$?

line 82:
why is the covariance matrix $\boldsymbol{\Sigma}_{i_1,...,i_d}$ equal to a scalar?

I mean in $\boldsymbol{\Sigma}_{i_1,...,i_d}=1$.

line 90: What is $1_{\nu_{\gamma}+1}$? The term was not introduced.

Should it be $(A_{\gamma})_{i,j}$? It is not clear how both equations in (10) are related.

line 91:
How is $\zeta_{\gamma}(i_{\gamma})$ related to equation (10)? It is not clear.

line 95:
The statement: "This adjustment works up to $\nu_{\gamma}= 25$, after which negative values occur." is an empirical fact? or is there any mathematical explanation for such a fact? Does it happen regardless of the dimensionality?

line 104 (Figure 2):
Typo, get rid of "is" in "the variance is of $f$ is more..."

---The comments below will appear separated a bit strange, but it is due to a problem in the reviewing platform---

line 108 :
It is a bit confusing to write, $\mathbf{P}_{i_1,...,i_d}$ following a Gaussian with

the mean $\boldsymbol{\hat{\vartheta}}_{i_1,...,i_d}$

and covariance $\boldsymbol{\hat{\Sigma}}_{i_1,...,i_d}$

since it was already defined for the prior, $\mathbf{P}_{i_1,...,i_d}$ (after Eq. (7)) to follow a Gaussian with

the mean $\boldsymbol{\vartheta}_{i_1,...,i_d}$

and covariance $\boldsymbol{\Sigma}_{i_1,...,i_d}$.

Actually for the "variational control points" it should be something like $\mathbf{P}_{i_1,...,i_d}|\mathbf{y}$ following a Gaussian with

the mean $\boldsymbol{\hat{\vartheta}}_{i_1,...,i_d}$

and covariance $\boldsymbol{\hat{\Sigma}}_{i_1,...,i_d}$,

given that it is the (approximate) posterior. Nonetheless, I suggest to define in terms of each variational distribution $q(\mathbf{P}_{i_1,...,i_d})$ equal to  a Gaussian with

the mean $\boldsymbol{\hat{\vartheta}}_{i_1,...,i_d}$

and covariance $\boldsymbol{\hat{\Sigma}}_{i_1,...,i_d}$,

this in order to ease the understanding of the following equations (15) and (16).

Specially to differentiate the prior $p(\mathbf{P}_{i_1,...,i_d})$

from the approximate posterior $q(\mathbf{P}_{i_1,...,i_d})$.

line 111:
How is $q(\mathbf{P})$ related to each $q(\mathbf{P}_{i_1,...,i_d})$? It need to be defined.

lines 110, 113 and 120:
the notation for the variable $y$ should be bold, I mean a vector or set that stacks all the $y_j$, say $\mathbf{y}$={$y_j$}$^n_{j=0}$.

line 116:
Why is it mentioned a variable $\sigma^2$ if the likelihood has not even been introduced?

line 120 Eq. (15):
There should be big brackets containing all the terms of the summatory.

line 127:
It might be better to make Figures appear after they are mentioned in the text.

line 151:
please introduce the variables $1_{\nu_{1}+1}^{\top}$ and $1_{\nu_{d}+1}^{\top}$.

line 225:
correct wording in "..., to gives insight into when..."

line 243:
Typo? include "be" in "..., $f$ should in the region..."

line 256:
introduce acronym SGPR.

lines 267 to 269:
It might be useful to know what are the results for keggdirected and elevators, but guaranteeing that the test points are not outside of the domain for the splits.

line 274:
The authors explain that for the concrete dataset the variance overfits yielding a non-optimal likelihood, and conjecture this might be due to $n/d$-radio being low. Nonetheless, the $n/d$-radio for energy is not far from concrete's, and we see from the Table 1 that the energy results were much better than SGPR. How did the authors come up with such a conjecture?

line 325: please upper case names in the references like: Gaussian, Bayesian, Bernstein,

---Other Questions---

I do not remember having seeing it in the paper, how are the predictions of the model performed? How does the computational complexity of the predictions differ from the classical SGPR?

An interesting property of a Sparse GP is that when the number of inducing points approaches to the number of data observations, we would expect to obtain almost the same results of a full (non-sparse) GP. Is this fact tested in the Bézier GP approach, I mean that when the number of control points approaches to the number of observations we would almost end up with the original Non-sparse GP?

Also, if we set the inducing points as exactly the same to the original input observations, we would expect to recover the original full (non-sparse) GP, is it possible to think about the Bézier GP approach as able to recover the full GP where the input data observations are the same control points?

Would it be feasible to think about a Deep Bézier Gaussian Process? What could it be the challenges and limitations?

The construction of the GP allow the use of different orders $\nu_d$ per dimension, when might that case of having different orders per dimension be useful or applicable?

**Limitations:**

The authors were critic about the limitations of their work.

**Strengths And Weaknesses:**

Originality: The work seems original and broadly different to the classical sparse Gaussian process approaches.

Quality: The experiments provided in the paper can be useful for practitioners in the field of Gaussian process models and can open an alternative path for researching about the scalability of Gaussian process models.

Clarity: The work is generally well written and not difficult to follow, though it presents some problems in the mathematical notation.

Significance (importance): Nowadays the application of Gaussian process models in the context of large datasets with a wide number of features in the input space is still limited due to scalability problem. Therefore, alternative GP approaches that aim to enhance the applicability of GP in such contexts of large and wide data are of great value for the research community.

---

> ### Author Response · Authors · 2022-08-01
> **Reviewer mrad**
>
> We thank the reviewer for all comments about typos, not all are commented upon here, but they have been read - and they will be fixed. Thank you for improving the quality of our paper.
>
> * Bold-facing
>
> We are aware of a slightly unconventional use of bold-facing when several places we only consider 1-d vectors (in other words, scalars). Our reasoning was that in principle the model formulation extends to higher dimensional outputs. This could make the model useful for latent variable models etc. in the future. We will ensure that we formulate this unusual convention at the very beginning of Section 2 to make sure readers know.
> We could have presented the framework for higher dimensional output from the start, but we felt this was an unnecessary complication as our focus for this paper was scalable GP regression. The extension to higher dimensional outputs is quite straightforward, but notationally a bit tedious.
>
> * Line 72
>
> This was to emphasise that now P is a random variable and not a parameter. We shall make this clear. We only consider P to be scalar, but as remarked above, it could in principle be a vector, therefore we boldface.
>
> * Line 90
>
> 1_nu refers to a vector of size nu with 1 in all entries. We will make sure this is clear in the manuscript.
>
> * Line 95
>
> We have no mathematical explanation for this fact, we simply observed it happened. It is independent of dimension, as Equation 10 is computed for each dimension separately.
>
> * Line 116
>
> Thank you for pointing this out. We shall clarify what sigma is here.
>
> * Line 267
>
> If we normalise the data, such that no test point fall outside the hypercube, we get an (average) RMSE of 0.0937 and log-likehood of 0.9601. Just marginally (but significant) better than SGPR. We will of course add this to the explanation of the results on this dataset. We thank the reviewer for this suggestion to improve our paper.
>
> * Line 274
>
> It is true that on energy we outperform SGPR. However, we still see overfitting issues. This is because the train and test RMSE are significantly different, on energy the train RMSE is around 0.04 for BezierGP (varying a bit with the order). In comparison, the train RMSE for SGPR is around 0.35. So the train fit is much better than the test fit for BezierGP. So yes, it is a better test fit than SGPR, but overfitting is still there for low n/d-ratio.
>
> * Prediction complexity
>
> The predictions are made by a simple forward pass in the Bezier buttress. This means complexity of O(nrd (nu)^2). See Line 186-189 for a description. In comparison, SGPR use complexity O(nm^2), where m is the number of inducing variables. However, our advantage is that we can add *many* control points in high-dimensional spaces and remain computationally feasible. It would be infeasible for SGPR to include enough inducing variables in very high dimensional spaces.
>
> * "Convergence" towards full GP
>
> In the BezierGP we are not free to place inducing locations, they are restricted to a grid. So there is no equivalent results to the one you mention. However, there is an analogous one: if we have nu + 1 data points we can infer the "true" polynomial GP. Assuming noise free and 1-dimensional input. The number of data-points needed scale exponentially with input dimensions.
> The full GP in our setting would be to compute the true posterior of the control points. We are for computational reasons restricted to make a mean-field variational approximation.
>
> * Deep Bezier GPs
>
> It would indeed be feasible to construct Deep Bezier GPs. In simple words it would be sequentially making forward passes through Bezier buttresses. Notice here that it would be necessary with high-dimensional outputs, which is possible, as we also state above. As we see it, the difficulty with deep Bezier GPs is the same as it is with Deep GPs in general: non-reversible non-linearities in the intermediate layers. See the thesis from David Duvenaud for further details on this.
> It is also unclear if deepness is advantegous as a chain of polynomials is again a polynomial. Hence it could be more interesting for future research to try and increase the order as an alternative.
>
> * Different orders per dimension
>
> We believe this one to potentially have a big impact. This is particularly in the case where some features are binary (or other finite discrete features). Here we could set the order of that feature to be 1 (effectively giving that feature 2 control points to vary), which would make it exact without unnecessary extra computation. This makes BezierGP also very applicable for inputs that are not Euclidean in nature.
> Also, we hope future work will try to optimise over orders for each dimension potentially finding features that are, say, linear. This should, hopefully, improve upon generalisation.

---

> > ### Comment · Reviewer_mrad · 2022-08-08
> > **Thanks for the Responses**
> >
> > I want to thank the authors of the paper for their effort to clarify my doubts and questions, I am happy with the different responses. I am just curious to see how the paper looks after correcting all the typos and the additional comments regarding future works and clarifications.
> >
> > In general I appreciate the effort to make the work of a high quality.
> >
> > Best,

---

### Official Review · Reviewer_rsfL · 2022-07-13

**Rating:** 5
**Confidence:** 3
**Soundness:** 3 good
**Presentation:** 3 good
**Contribution:** 3 good

**Summary:**

The manuscript proposes a methodology to extend the Gaussian process framework towards datasets that are both high-dimensional (wide) and have a large sample size (tall) using multivariate Bézier curves. In particular, the authors propose to perform variational inference over a multivariate linear model over Bernstein polynomials having exponentially many control points as parameters. To render the model scalable, the authors propose two approximations: They use a factorisation assumption to reduce the number of control point parameters and they neglect the order of the summation over the dimensions. Experiments on several UCI benchmarks conclude the paper.

**Questions:**

* Computations with Bernstein polynomials usually require special care regarding numerical stability i.e. De Casteljau's algorithm. Do you encounter issues in that respect?
* The "adjustment" of the Bernstein polynomials might be numerically challenging as shown by Figure 2 right Panel. Is this correct? What is the condition of the linear system in Equation (10)?
* How would experimental results change without that correction?
* I'm missing an analysis and more detailed evaluation on the effects of the two approximations made to render the model scalable: the factorisation structure and the order of summation.
* In particular, how is the marginal likelihood bound affected? Is it still a bound or a mere approximation that cannot be trusted as a comparative measure in the experiments? Columns 2,4,6 of Table 2 suggests exactly that behavior. Then, one could only compare RMSE and not probabilistic model properties. More specifically, the comparisons in Figure 5, in the left panels and Table upper part would be questionnable in their meaning.
* What would need to change if the likelihood was non-Gaussian? I'm assuming that the same construction can be used.

**Limitations:**

Yes.

**Strengths And Weaknesses:**

1) Clarity
The paper is mostly well written and the technical content is rather well accessible.

2) Originality
The proposed polynomial covariance structure using Bernstein polynomials seems to not have been explored before.

3) Significance
The method is (currently) specific to large scale regression and could be useful for GP regression. However, in the current state, the underlying approximation assumption need to be better analysed before the model can be used as a black box.

4) Reproducibility
Unfortunately, only some vague outline of the code is contained in the appendix suggesting that there might be an implementation that might be shared at some point. The relevant classes i.e. BezierButtress are not contained hence this cannot be judged. Assuming that the code will be shared, the experiments should be well reproducible. If not, this should be rather difficult.

5) Empirical analysis
The manuscript contains experiments on several standard regression datasets regarding marginal likelihood and prediction error.
a large number of numerical results gathered in a big table regarding marginal likelihood, prediction error and runtime. Some more results on more subtle aspects of the covariance function itsel would have been much appreciated.

---

> ### Author Response · Authors · 2022-08-01
> **Reviewer rsfL**
>
> Thank you for your insight and improving the quality of our work. We hope that we adequately addressed your questions, and that you would consider the replies in your final evaluation of the paper. We’d be happy to comment further. Also thank you for asking questions we had not considered, this improves the paper, we hope the reviewer thinks so too.
>
> * Code (BezierButtress)
>
> The code for the class BezierButtress is contained in controlpoints.py which was in the supplementary material. Can the reviewer please check that it is indeed there?
> The reviewer is correct that only the most substantial parts of the code have been added to the supplements, and the reviewer is too correct that the full (unanonymised) code will be released upon acceptance.
>
> * Numerical stability (De Casteljau)
>
> We did not encounter any numerical instabilities associated with the Bernstein polynomials. We are aware that such could exist for higher orders (we only went to 20). We will remark this in the paper as a potential caution to future readers.
>
> * Bernstein adjusting (stability)
>
> Yes, it is very correct. The current adjustment is only accurate up to order 25, after this it returns negative variance values, which of course is not permitted. We did note this in the paper.
>
> * How do Bernstein adjustment change results?
>
> We performed experiments on the bike and protein dataset without the Bernstein correction. Surprisingly, in these experiments, the performance was not significantly worse than with the prior adjustment. This is promising as it opens the path to do Bezier GPs with higher orders than 25 (see above).
>
> * Analysis on the two approximations
>
> We agree with the reviewer that a full mathematical analysis of the parametrisation is warranted. What we can say at this time is that for 1 and 2 dimensional inputs the current parametrisation has the same "capacity" as free parametrisation. By free parametrisation we mean one parameter per control point, ie. no factorisation structure. For 3 dimensional input if we include all orderings the same holds. We are working on a deeper and more general framework to proof that for any dimension the factorisation structure have same capacity as the free parametrisation if we include all orderings of the dimensions.
> This will hopefully give more insights into how many permutations (orderings) are needed for good approximation of the free parametrisation.
> For this rebuttal we have deepened the empirical analysis for Table 2 so the optimisation noise is discounted (see point below).
>
> * How is bound affected by the approximations, is bound exact?
>
> Thank you for this question, which we had not considered but should! The ELBO we present is still a lower bound to the marginal likelihood. The use of Jensen's inequality is independent of the choice of parametrisation, and hence choice of our factorisation and summation. Of course, the choice of parametrisation can obstruct the model of getting close to the optimal value of the ELBO, but we think the results emphasise that this would not be the case. Therefore, the probabilistic model properties can be trusted, if we find a good ELBO value.
> The numbers in Table 2 are a bit noisy due to optimisation effects, we will update this table with multiple restarts to get a clearer picture.
>
> * Non-Gaussian likelihoods
>
> You are correct that the same construction can be used for non-Gaussian likelihoods. One issue that would arise would be to find a optimisation routine that accounts for large KL's in the initial phase (we fix likelihood variance small in the initial phase). Yet, as also pointed out by reviewer W66W, this is effectively a KL annealing scheme, so there is reason to believe it should work for other likelihoods too. Mathematically, nothing hinders any other choice of likelihood.

---

### Official Review · Reviewer_W66W · 2022-07-15

**Rating:** 5
**Confidence:** 4
**Soundness:** 3 good
**Presentation:** 4 excellent
**Contribution:** 3 good

**Summary:**

The paper presents a Gaussian process method focusing on data in which the number of samples and the number of dimensions are both high. The GP prior is built upon Bezier surfaces and is formulated as a combination of Gaussian random variables and coefficients taken from Bernstein polynomials. The paper suggested adjusted priors and scalable construction via Bezier buttress. Experiments are performed on various data sets.

**Questions:**

- How to perform the matrix inverse of $k(Z,Z)$? Is it done analytically by using inverse squared Bernstein.  If so, I would suggest to put more explanation or reference for inverse squared Bernstein and Bernstein polynomials which I feel is missing in this paper. (I’m not familiar with the literature here).

- The construction of $f(x)$ somewhat resembles deep structures (of course with no activations), the comparison can be done extensively with other models like deep Gaussian process or deep kernel learning. However, the paper reports just one result with DGP for the “year” data set. Are the results for the remaining data sets (buzz, house electric, slice) available?

Minor points:
	-The optimization is somehow pragmatic. Since the paper suggests KL terms may be big at initial steps, I think KL annealing techniques can be used here.


**Ethics Review Area:**

["I don’t know"]

**Limitations:**

Please see above.

**Strengths And Weaknesses:**

Strengths:

- The problem of handling high-dim GP is interesting , the approach is novel to tackle the problem in GP research.
- The paper gives a good analysis for the method and provides a scalable solution for GP.

Weakness:
- I have one concern of how well Bezier d-surfaces can approximate data. What is the error as the number of dimensions increases? If there is any reference for this, it would be nice to add the clarification.
- It might be out of scope in this paper, there is no theoretical justification for that the proposed approach can address effectively high-dim data .This may be related to the previous point and the way the paper suggests constructing Bezier buttress. The empirical results (first plot of Fig 5.) may. The main concern is that we do not know whether the ability to handle high-dim data comes from deep networks or from the proposed approach.

---

> ### Author Response · Authors · 2022-08-01
> **Reviewer W66W**
>
> Thank you for your insight and for improving the quality of our work. We hope that we have adequately addressed your questions. We’d be happy to comment further if there are remaining points!
>
> * How well can Bezier d-surfaces approximate data in high dimensions
>
> It is true for any polynomial regression that with o^d + 1 observations (assuming noise-free), we can retrieve the true function (if the true function is a polynomial of that order). Further Bernstein polynomials are universal function approximators in any dimension (see Hildebrandt reference) when the order goes to infinity. So Bezier d-surfaces can too approximate any continuous function if the order is of adequate size. Of course, the question of how the error grows with the dimensionality is the interesting one. This is of course very dependent on the nature of the true function and the (possible) Lipschitz constraints one can make.
>
> * Theoretical guarantees (and where does the ability to handle high dimensions come from)
>
> To the question of theoretical guarantees we refer to the above. There are theoretical guarantees that Bezier d-surfaces can approximate any continuous function on [0,1]^d.
> The ability to learn high-dimensional data in this paper does not come from any deep structure. The deep structure (the Bezier buttress) is actually a limiting construction on the learning ability as it limits the parametrisation of the control points. The deep structure (the buttress) is there to make it scalable so that we can infer many control points.
>
> * Inversion of k(Z,Z) and about the inverse squared Bernstein
>
> In fact, there is never any inversion of k(Z,Z). A major feature of our approach is that we never have to invert matrices or compute determinants during GP inference or prediction. Both inference and prediction is performed as a forward pass in the Bezier buttress.
> The inverse squared Bernstein is only used as a one-off computation to adjust the prior variance to be similar to common GP priors. As such, it has nothing to do with inversion of k(Z,Z) (or others). The term 'inverse squared Bernstein' is our own term, so we understand that the reviewer is not familiar with the literature on it. We will make this more apparent in the manuscript.
>
> * Resemblance with deep structures (other results?)
>
> The deep structure in the references you mention are different from our deep structure. Both DGPs and Deep Kernel Learning rely on (sequential) feature learning, whereas our deep structure is *not* a feature learner, but simply an instrument we use to sum over many control points. However, it is possible that in the future BezierGPs can be stacked to form Deep GPs.
> On the question, if there are results on DGP on the three datasets you mention, we have not come across them.
>
> * KL-annealing
>
> We are happy that the reviewers see potential in KL-annealing as an alternative to our optimisation routine. We share this perspective. In fact, fixing the likelihood variance to be very small is effectively a KL-annealing as it amplifies the likelihood term in the beginning of training. But more advanced annealing routines could be tested.

---

### Meta-Review · Area_Chair_jDpx · 2022-08-29

**Recommendation:** Accept
**Confidence:** Certain

**Metareview:**

The submission considers Bernstein polynomial-based kernels for large-scale Gaussian process regression. To deal with high dimensional inputs, two approximations were proposed: a parameterisation to amortise/share parameters between the random control points of the polynomials and ensembling over randomly permuted orders. In addition, a prior adjustment to the polynomial coefficients was introduced to retain the desirable properties of the variance in the prior.

The reviewers raise some concerns about these approximations in terms of approximation quality and impact on predictive performance, and the overfitting issue. The authors acknowledged some of these concerns in the text and during the rebuttal.  The AC also notes that the "behaviour foundational to practitioners" [Line 250] is not illustrated well, and perhaps a comparison to non-Bernstein polynomial kernels could be provided to demonstrate the differences/benefits.

Overall, all reviewers think the proposed method is interesting and offers a fairly novel approach to scalable GP regression with high dimensional inputs (with some limitations that require further experiments/analyses), and thus the paper should be accepted. We hope the comments/discussions here are useful for the next iteration.

**Award:**

No

---

### Decision · Program_Chairs · 2022-09-14

Accept